# A Model between Near-Surface Air Temperature Change and Dynamic Influencing Factors in the Eastern Tibetan Plateau, China

**DOI:** 10.3390/s22166196

**Published:** 2022-08-18

**Authors:** Wentao Xu, Qinjun Wang, Dingkun Chang, Jingjing Xie, Jingyi Yang

**Affiliations:** 1Key Laboratory of Digital Earth Science, Aerospace Information Research Institute, Chinese Academy of Sciences, Beijing 100094, China; 2International Research Center of Big Data for Sustainable Development Goals, Beijing 100094, China; 3Yanqihu Campus, University of Chinese Academy of Sciences, Beijing 101408, China; 4Key Laboratory of the Earth Observation of Hainan Province, Hainan Aerospace Information Research Institute, Sanya 572029, China

**Keywords:** temperature change, influencing factors, eastern Tibetan Plateau (TP), statistical analysis, correlation coefficient

## Abstract

Climate change, characterized by global warming, is profoundly affecting the global environment, politics, economy, and social security. Finding the main causes of climate change and determining their quantitative contributions are key points to making climate decisions on responses to climate change. The Tibetan Plateau (TP) is sensitive to global climate change. Taking the 100 km buffer zones of 45 meteorological stations in the eastern TP as research objects, we conducted an experimental study on temperature change and its influencing factors. Using the least squares multivariate statistical analysis method, a model between the annual and seasonal standardized temperature change and its dynamic influencing factors in the past 20 years was established. The results showed that, in the eastern TP, temperature change was affected by different factors in different periods. Vegetation cover and snow cover were the most correlated factors to temperature change. The influence of carbon dioxide, vegetation cover, and water cover was subject to seasonal changes. Urban cover and bare land cover did not pass the *t*-test. This research not only provides a theoretical basis for the analysis of temperature change over the TP, but also points out the direction for the analysis of temperature change causes in three polar regions.

## 1. Introduction

As one of the most severe challenges facing the world, climate change, characterized by global warming, is influencing the ecological environment and social development. According to the “State of the Global Climate 2020” report released by the World Meteorological Organization (WMO), the period from 2011 to 2020 was the warmest decade in the global instrumental record. The direct impact of climate change is the frequent occurrence of extreme weather and climate events, changes in global glaciers and snow caps, sudden geological disasters, etc., which further affects the living environment and development of human beings.

Due to the sensitivity of the Tibetan Plateau (TP) to global climate change, many scholars have conducted research on climate elements of TP and its surrounding areas in recent decades. Global glaciers have been projected to lose 18–36% of their 2015 ice volume by the end of the 21st century due to many primary drivers, such as air temperature, snowfall, rainfall [1]. Especially, some scholars have analyzed the spatiotemporal patterns of China’s glaciers in the past half-century, and pointed out that, due to climate changes and local scale factors, glaciers on the TP retreated from the northwest to southeast, with a high mass loss of glaciers in the southeastern TP [2,3,4,5]. The extreme precipitation and temperature caused by climate change leads to frequent disasters. The evaporation of soil moisture in southwestern China has increased due to temperature increase, which leads to the decrease of slope stability. Combined with the increase of precipitation, the probability of geological disasters, such as glacier debris flow and mountain landslides, has also increased [6,7]. Furthermore, the spatiotemporal changes of climate change factors such as temperature and precipitation directly affect the habitat conditions of plants growth and ecosystem productivity. Many dynamic vegetation models and methods have been used to evaluate the influence of climate change on the ecosystem of the TP [8,9]. For example, Shen et al. used the Maximum Enhanced Vegetation Index (MEVI) to analyze the vegetation cover changes in the growing period of the TP from 2000 to 2012 [10]. Some researchers analyzed the spatiotemporal variation trends of the NDVI and its relationship with precipitation and temperature over the TP in the past 40 years [11,12].

It can be seen that most of the current research is limited to the spatial-temporal changes and responses of environmental elements to climate change in the Plateau. However, there are few studies on the quantitative analysis of the influencing factors of temperature change in the eastern TP, especially dynamic influencing factors, and no quantitative model has been established between temperature change and its dynamic influencing factors. Therefore, establishing a quantitative model between temperature change and its dynamic influencing factors can not only discover the principles of the temperature change in the TP, but also quantitatively determine the contribution of different factors to temperature change, which is of great significance for world climate research and national climate strategy. The premise hypothesis of this research is that the temperature change in the eastern TP in different periods (annual/seasonal) is affected by all dynamic influencing factors selected in this article. The spatial heterogeneity was fully considered and the extraction of the meteorological station data was according to the geographical law of “All objects are related, but objects with close distance are more related than those with far distance”.

Studies [13,14,15,16] show that temperature change is influenced by static factors such as geographical location, altitude, solar activities, and other natural factors, as well as dynamic influencing factors, such as atmospheric dynamics, land covers (vegetation, urbanization, snow, etc.), and greenhouse gas concentrations.

The temperature in this article refers to the daily average air temperature, which is averaged for temperature measurements at multiple time points using thermometer screen with 1.5 m above the ground. Its spatial-temporal variations are important to evaluate the amplitude, rate, and variation characteristics of local and global temperature changes [17,18,19,20].

## 2. Materials and Methods

### 2.1. Study Area

The TP is the largest plain in China and the highest in the world, with the Himalayas in the south, the Kunlun Mountains, the Altun Mountains and the northern edge of the Qilian Mountains in the north, the Pamir Plateau and the Karakoram Mountains in the west, and the western part of the Qinling Mountains and the Loess Plateau in the east. With an area of $25,724\times 10^{3}\;{\mathrm{km}}^{2}$ and average altitude of 4000 m a.s.l, it is well known as “the Third Pole”. It has rich natural landscapes controlled by different weather systems [21,22].

Due to the scarcity of meteorological stations in the western TP, the study area in this article is mainly located in the eastern TP whose elevation is from 440 m to 8500 m a.s.l. With the geographic coordinates range of 85°10′–104°40′ E, 25°50′–39°46′ N, the highest altitude of the meteorological station is about 4720 m a.s.l, and the lowest is about 2380 m a.s.l. Although the average annual (seasonal) temperature of the TP is relatively low, it has shown a rising trend in recent decades; the temperature change of eastern TP from 1980 to 2021 is shown in Figure 1. In addition, the average carbon dioxide concentration of the eastern TP has also increased rapidly in the past two decades (Figure 2). The Digital Elevation Model (DEM) derived from the Dataset of Chinese 1 km resolution Digital elevation Model (National Cryosphere Desert Data Center, Lanzhou, China) and the distribution of 45 meteorological stations in the article are shown in Figure 3. It is necessary to emphasize that the distribution of meteorological station buffer basically covers most of the eastern TP, so the affected area in this article is almost the entire eastern plateau.

### 2.2. Data

#### 2.2.1. Data Acquisition

In this research, data on temperature and its dynamic influencing factors, such as carbon dioxide concentration, vegetation cover, urban cover, snow cover, water cover, and bare land cover were collected.

Carbon dioxide in the atmosphere can pass through the Sun’s short-wave radiation and absorb long-wave radiation emitted by the Sun and the Earth’s surface, allowing only a small amount of thermal radiation to be lost to space. The atmosphere absorbs more radiant heat than it loses, so air temperature rises as carbon dioxide increases [23].

The reflectance of various vegetation is different, and the reflectance of vegetation covered area is much smaller than that of bare land. Through the change of underlying surface and photosynthesis, the solar radiation and heat balance change, thus leading to the change of the air temperature. Generally speaking, vegetation has the effect of cooling and reducing humidity. The Normalized Difference Vegetation Index (NDVI) can dynamically monitor vegetation cover and growth at large spatial-temporal scales through the combined operation of infrared and near-infrared bands [12].

Temperature change caused by urban cover is mainly through the heat island effect [24,25], the greenhouse effect caused by human activities, and the parasol effect of increased soot. Among them, the heat island effect and the greenhouse effect will increase the air temperature. The soot suspended in the atmosphere can reflect solar radiation and condense water vapor, therefore, it will weaken the solar radiation and cause temperature reduction. Nighttime light is an important indicator of human activities, including social economy and energy consumption, which has been widely used in urban area identification, etc. [26].

Snow cover is an important part of the cryosphere, which is of great significance to the study of land surface energy and water exchange, mountain hydrology, etc. [27]. The surface of ice and snow has a high reflectance to the Sun radiation, and it can absorb little solar radiation, so it has a weak exchange capacity with the atmosphere. In addition, snow has a cooling effect because it can block the transfer of heat from the surface to the atmosphere.

Water can both absorb and release heat, so the temperature changes with the hydrologic cycle [28]. Scholars show that the surface reflectance of bare land has a certain influence on air temperature [29]. However, the extent to which they affect temperature is unclear.

We have obtained the data of meteorological stations in the study area, the reanalysis data of carbon dioxide concentration, and the land cover datasets of MODIS, Landsat, and other satellite inversion products. Shown in Table 1, all data are from 2003 to 2020.

Among them, the measured meteorological element data of meteorological stations provided by NOAA are international exchange data with a climate exchange agreement called Resolution 40 (Cg-XII) from the World Weather Observation Program of the United Nations WMO; the Copernicus Atmospheric Monitoring Service’s (CAMS) global greenhouse gas reanalysis (EGG4) monthly averaged fields dataset is part of the European Center for Medium Weather Forecasting (ECMWF) atmospheric composition reanalysis focusing on long-lived greenhouse gases [30]; the MOD13A3 of the LP DAAC Data Center is a monthly gridded level 3 product in the sinusoidal projection; A Prolonged Artificial Nighttime-light Dataset of China (1984–2020) from National Tibetan Plateau Data Center is generated by using a Night-Time Light convolutional Long Short-Term Memory (NTLSTM) network. The results can capture the time trend of the newly built-up areas well and show the road networks; daily cloud-free snow cover products for Tibetan Plateau from 2002 to 2021 not only fills in the data gaps caused by frequent clouds, but also improves the accuracy of the original MODIS snow products. In the case of snow transition period, complex terrain with high altitude and sunny slope, the accuracy of new snow products is higher than that of original data [31].

#### 2.2.2. Data Preprocessing

(1) Due to the different spatial resolutions of different data types, data should be resampled.

(2) According to the daily surface meteorological observations of 45 meteorological stations in the eastern TP from 2003 to 2020, the monthly average air temperature of each station was obtained by averaging daily temperature during each month.

The seasonal air temperature is obtained by further classifying the monthly average temperature. This article adopted the accepted method of seasonal division in climatology, the average temperature from March to May is taken as the spring average temperature, that from June to August as the summer average temperature, that from September to November as the autumn average temperature, and that from December to the February in the next year as the winter average temperature [11].

(3) Extracting the corresponding data of various dynamic influencing factors, such as carbon dioxide concentration, land use of various types (vegetation, snow, water, urban, bare land), by setting appropriate buffer zone of stations.

The regional carbon dioxide concentration data of each meteorological station is approximately considered as the carbon dioxide concentration of the corresponding station.

Vegetation cover and bare land cover are the corresponding areas in the buffer zone of each meteorological station extracted by NDVI. According to previous experience, the part of NDVI index between 0 and 0.1 is approximately considered as bare land. Vegetation area is considered to be the part of NDVI index greater than 0.1. Although there is a distinction between sparse and dense vegetation, it is not subdivided in this article.

Urban cover is the urban area within the buffer zone of each station extracted by nighttime light intensity; snow cover refers to the extraction of snow-covered area within buffer zones.

As for water cover, the high reflectance parts with NDVI index less than 0 need to be first extracted, and then the water cover in different seasons is obtained by interpolation between the high reflectance area and snow cover area obtained from snow products. This method can take into account the change of land cover in four seasons to a certain extent.

(4) Due to the different time scales, for example, snow cover dataset is daily scale, vegetation cover is monthly scale, so it is necessary to use a similar way as seasonal division to get seasonal (annual) scale data of each data type.

(5) Due to the different properties of dynamic influencing factors, they usually have different magnitude and dimensions. When the order of magnitude of each factor varies greatly, the effect of parameters with higher value will be highlighted in the comprehensive analysis, while that with lower will be weakened [32]. Therefore, it is necessary to standardize the dynamic influencing factors.

In this way, the static influence of changes from natural conditions on local temperature is eliminated to a certain extent because changes in static influencing factors (such as the Sun, altitude, location, topography, etc.) had basically the same contribution to temperature change in the past 20 years in a same station.

The related definitions are as follows.

Anomaly

Anomaly is used to indicate the deviation of a factor from the normal level, which can measure the fluctuation of factors and eliminate the influence of natural conditions on samples to a certain extent [33]. It can be expressed by
(1)$$X_{i}=x_{i}-\bar{x}\;,\;\;\;\;\;\;\;i=1,2,3\cdots \;n$$
in which, $x_{i}$ is the observed factor at a meteorological station in *i*th year; $\bar{x}$ is the average value of this factor at this station for n years; $X_{i}$ is the anomaly value of the factor in the *i*th year at the station.

2.Data standardization

The standardized value can be expressed by
(2)$$Z_{i}=(x_{i}-\bar{x})/s_{i}\times 100\%,\;\;\;\;\;\;\;i=1,2,3\cdots \;n\;$$
in which, $x_{i}$ is the observed factor at a meteorological station in *i*th year; $\bar{x}$ is the average value of this factor at this station for n years; $s_{i}$ is the standard deviation of this factor at this station for years; $Z_{i}$ is the standardized data of the factor in the *i*th year at this station. By means of this method, it can not only make full use of the variation characteristics, but also eliminate the problem of the large difference of each factor in different buffer values.

### 2.3. Methods

#### 2.3.1. Multiple Regression Analysis Method

Multiple regression analysis is a statistical analysis method, whose principle is to establish the relationship model between dependent variable and independent variables.

Its formula can be expressed by
(3)$$Y=\beta_{0}+\beta_{1}X_{1}+\beta_{2}X_{2}+\cdots +\beta_{k}X_{k}+\varepsilon$$
in which, Y is the dependent variable; $X_{k}$ is the *k*th independent variable, and $\beta_{k}$ is the *k*th regression parameter, ε is the residuals.

Taking the mathematical expectations on both sides of Equation (3), the overall regression equation is obtained as
(4)$$E\left(Y|X_{1},X_{2},\cdots ,X_{k}\right)=\beta_{0}+\beta_{1}X_{1}+\beta_{2}X_{2}+\cdots +\beta_{k}X_{k}$$
in which, $E(Y|X_{1},X_{2},\cdots ,X_{k})$ represents the mean of the observed value Y under the condition of the given independent variable $X_{i}$. Since $\beta_{0},\beta_{1},\beta_{2},\cdots ,\beta_{k}$ are unknown, the corresponding estimates of the overall parameters $\hat{\beta_{0}},\hat{\beta_{1}},\hat{\beta_{2}},\cdots ,\hat{\beta_{k}}$ need to be given according to the sample observations, and the sample regression equation is obtained as
(5)$$\hat{Y}=\hat{\beta_{0}}+\hat{\beta_{1}}X_{1}+\hat{\beta_{2}}X_{2}+\cdots +\hat{\beta_{k}}X_{k},$$
in which, $\hat{Y}$ is the estimation of $E(Y|X_{1},X_{2},\cdots ,X_{k})$.

Estimated parameters can be obtained by minimizing the least square estimations
(6)$$Q=\sum {\left(Y_{i}-\hat{Y_{i}}\right)}^{2}=\sum {\left(Y_{i}-\hat{\beta_{0}}-\hat{\beta_{1}}X_{1}-\hat{\beta_{2}}X_{2}-\cdots -\hat{\beta_{k}}X_{k}\right)}^{2}$$

According to above formula, the estimated values of the parameters $\hat{\beta_{0}},\hat{\beta_{1}},\hat{\beta_{2}},\cdots ,\hat{\beta_{k}}$ can be solved by taking the partial derivative of *Q* with respect to $\hat{\beta_{0}},\hat{\beta_{1}},\hat{\beta_{2}},\cdots ,\hat{\beta_{k}}$ and making it to be 0.

#### 2.3.2. Evaluation indicators

(1) Pearson correlation coefficient

Pearson correlation coefficient is a value that indicates the degree of linear correlation between two variables [34], which can be expressed by
(7)$$r\left(X,Y\right)=\frac{\sum_{i=1}^{n}\left(X_{i}-\bar{X}\right)\left(Y_{i}-\bar{Y}\right)}{\sqrt{\sum_{i=1}^{n}{\left(X_{i}-\bar{X}\right)}^{2}}\sqrt{\sum_{i=1}^{n}{\left(Y_{i}-\bar{Y}\right)}^{2}}}$$
in which, r(X,Y) is the correlation coefficient; n is the sample size; $\bar{X}$ and $\bar{Y}$ are mean values of variables X and Y, respectively. The closer the absolute value is to 1, the stronger the correlation becomes.

(2) Multi-correlation coefficient

Multi-correlation coefficient is an indicator that reflects the degree of correlation between a dependent variable and a group of independent variables [35]. The larger the multi-correlation coefficient is, the closer the correlation becomes. The process to find out the multi-correlation coefficient between a dependent variable Y and independent variables is as follows.

① Making regression of *y* and $x_{1},x_{2},\cdots ,x_{k}$,
(8)$$\hat{y}=\hat{\beta_{0}}+\hat{\beta_{1}}x_{1}+\cdots +\hat{\beta_{k}}x_{k}$$
in which, $x_{1},x_{2},\cdots ,x_{k}$ are variables, $\hat{\beta_{k}}$ is the regression coefficient of each variable, and $\hat{y}$ is the predicted value of the dependent variable.

② Calculation on the simple correlation coefficient between y and variables $x_{1},x_{2},\cdots ,x_{k}$,
(9)$$R=\frac{\sum_{i=1}^{n}\sum \left(y_{i}-\bar{y}\right)\left(\hat{y_{i}}-\bar{y}\right)}{\sqrt{\sum_{i=1}^{n}\sum {\left(y_{i}-\bar{y}\right)}^{2}{\left(\hat{y_{i}}-\bar{y}\right)}^{2}}}$$
in which, R is the multi-correlation coefficient; n is the sample size; $y_{i}$ is the actual value of the dependent variable; $\hat{y_{i}}$ is the predicted value of $y_{i}$, and $\bar{y}$ is the mean value of y. The closer the value is to 1, the stronger the correlation becomes.

(3) *T*-test and *p*-value

The *t*-test of multiple linear regression is the significant test of the coefficient of a single variable, usually measured by the *p*-value, which indicates the possibility of an event happening. The calculation method is as follows.

① Left side test $H_{0}$: $\mu \geq \mu_{0}$, $H_{1}$: $\mu <\mu_{0}$


The *p*-value is the probability that the test statistic is less than or equal to the test statistic value from the actual observed sample data when $\mu =\mu_{0}$.
(10)$$P-value=P\left(Z\leq Z_{c}|\mu =\mu_{0}\right)$$

② Right side test $H_{0}:\mu \leq \mu_{0}$, $H_{1}$: $\mu >\mu_{0}$


The *p*-value is the probability that the test statistic value is greater than or equal to the test statistic value from the actual observed sample data when $\mu =\mu_{0}$.
(11)$$P-value=P\left(Z\geq Z_{c}|\mu =\mu_{0}\right)$$

③ Two-side test $H_{0}$: $\mu =\mu_{0}$


The *p*-value is the probability that the test statistic value is greater than or equal to the absolute value from the actual observed sample data when $\mu =\mu_{0}$.
(12)$$p-value=2P(Z\geq |Z_{c}\left|\right|\mu =\mu_{0})$$
in which, Z represents the test statistic value, and $Z_{c}$ represents the test statistic value calculated from the sample data. Generally, *p* < 0.05 is considered to be statistically significant, which means that the probability of the difference between samples caused by sampling error is less than the significance level.

After the *p*-value is calculated, the given significance level α is compared with the *p*-value in the case that the null hypothesis is true. If α > *p*-value, reject the null hypothesis at the significance level α; otherwise, accept the null hypothesis.

#### 2.3.3. Technical Flowchart

In order to establish a model between the standardized temperature change and its dynamic influencing factors, firstly, it is necessary to determine the buffer size of the meteorological station. Then, using the single-factor analysis method to determine the most closely correlated factors in different periods (annual/seasonal) after data extraction. Finally, the least squares multivariate statistical analysis method is used to establish the model between the standardized temperature change and dynamic influencing factors in different periods.

##### Station Buffer Zone Selection

Some meteorological stations in the eastern TP are relatively close to each other, so setting the 150 km buffer zone will cause the overlapping of these stations; while setting the 50 km buffer zone will make most stations unable to be effectively connected. Therefore, setting a 100 km buffer zone for experimental research can not only reflect the independence of each station, but also ensure that most stations have certain connected areas.

##### Sensitive Factors Selection

In order to obtain the dynamic factors that mainly affect the temperature change in different periods (annual/seasonal), the single-factor correlation analysis is carried out according to the standardized data of them. Based on the principle of retaining data within three standard deviations and eliminating deviated abnormal data, the correlation coefficients between the standardized temperature change (t) and each factor in different periods can be obtained.

##### Model

Based on the above analysis, the least squares multivariate statistical analysis method is used to establish the model for the standardized temperature change (t) with selected dynamic influencing factors in different periods.

After establishing the model, it is decided whether to improve it or not based on its accuracy. When the accuracy satisfies the precision requirements, the model is output. If not, it is necessary to analyze the causes and re-establish the model. For example, if the *p*-value of the factor does not pass the *t*-test, it needs to be removed and re-established. Figure 4 is a technical flowchart of the model.

## 3. Results

The correlation coefficients between the standardized temperature change and six dynamic influencing factors selected in this article in different periods are shown in Table 2, Table 3, Table 4, Table 5 and Table 6, in which abnormal data were eliminated. There were 810 groups of original data in each season, and 778 groups, 801 groups, 761 groups, 778 groups, and 797 groups were analyzed for annum, spring, summer, autumn, and winter after eliminating outliers.

From Table 2, Table 3, Table 4, Table 5 and Table 6, the factors that passed the correlation test and had a correlation greater than 0.3 were selected for subsequent multiple regression analysis.

The model for the standardized temperature change (t) in different periods with selected dynamic influencing factors was established in which factors whose *p* values did not pass the *t*-test were removed.

①*Annum*: 

$$t_{1}=-0.02785+0.2258x_{1}+0.1511x_{2}-0.2562x_{3}\;$$
in which, $t_{1}$ is the standardized annual temperature change; $x_{1}$ is the standardized annual carbon dioxide concentration; $x_{2}$ is the standardized annual vegetation cover and $x_{3}$ is the standardized annual snow cover. Among them, the modeling factor standardized annual bare land cover did not pass the *t*-test and was removed.

②*Spring*:

$$t_{2}=-0.003515+0.3026x_{1}-0.3232x_{2}\;$$
in which, $t_{2}$ is the standardized spring temperature change; $x_{1}$ is the standardized spring vegetation cover; $x_{2}$ is the standardized spring snow cover. Among them, the modeling factor standardized spring bare land cover did not pass the *t*-test and was removed.

③*Summer*:

$$t_{3}=0.00565+0.1928x_{1}+0.09237x_{2}-0.3685x_{3}-0.2307x_{4}\;$$
in which, $t_{3}$ is the standardized summer temperature change; $x_{1}$ is the standardized summer carbon dioxide concentration; $x_{2}$ is the standardized summer vegetation cover; $x_{3}$ is the standardized summer snow cover and $x_{4}$ is the standardized summer water cover.

④*Autumn*:

$$t_{4}=0.013803+0.3582x_{1}+0.2931x_{2}-0.1637x_{3}\;$$
in which, $t_{4}$ is the standardized autumn temperature change; $x_{1}$ is the standardized autumn carbon dioxide concentration; $x_{2}$ is the standardized autumn vegetation cover and $x_{3}$ is the standardized autumn snow cover. Among them, the modeling factor standardized autumn bare land cover did not pass the *t*-test and was removed.

⑤*Winter*:

$$t_{5}=0.0002363+0.1843x_{1}-0.2846x_{2}\;$$
in which, $t_{5}$ is the standardized winter temperature change; $x_{1}$ is the standardized winter vegetation cover; $x_{2}$ is the standardized winter snow cover. Among them, the modeling factor standardized winter bare land cover did not pass the *t*-test and was removed.

The fitting relationship between the actual values and the predicted values of the model in different periods are shown in Figure 5.

As can be seen from Figure 5, the determination coefficient ($R^{2}$) of the model in annum, spring, summer, autumn, and winter are 0.2034, 0.2811, 0.2657, 0.3863, and 0.1711 respectively. The *t*-tests for each parameter in different periods are shown in Table 7, Table 8, Table 9, Table 10 and Table 11.

## 4. Discussion

Although carbon dioxide concentrations in winter and spring are relatively higher than that in spring and summer, we suspect the TP’s temperature change in these two periods affected by carbon dioxide concentration is not significant probably because the vegetation photosynthetic ability is weak and the carbon dioxide activity is low due to the low temperature [36,37]. Therefore, carbon dioxide concentration’s change in spring and winter is not as significant as that in summer and autumn. In addition, this feature can also be seen from the carbon dioxide tendency rates in each period in Figure 2, which are 20.35 ppm/10a, 20.22 ppm/10a, 20.90 ppm/10a, 20.25 ppm/10a, and 19.97 ppm/10a in annum, spring, summer, autumn, and winter, respectively.

Since various vegetation types have different sensitivity to temperature change, the impact of vegetation on temperature change in different periods is also different [12]. Some researchers analyzed the spatiotemporal variation trends of the NDVI and its relationship with precipitation and temperature over the TP over the past 20 years, the results showed that the overall NDVI of the TP had been increasing stably, which is consistent with our research conclusion. For example, Han et al. found that TP’s NDVI increased by 0.128 for every 1 °C increase in temperature because the net photosynthetic rate of plant leaves increased with temperature increasing [11]. Under the combined influence of temperature and precipitation, the vegetation cover in this region has a strong positive response to temperature. Under the condition of heat as the primary driving force, the vegetation growth in most areas has been significantly improved, and the area of vegetation improvement is greater than that of degradation, especially in the growing season, such as spring and autumn [12,38,39,40]. Liu and Chen et al. found that the prolongation of the growing season in the eastern and northern parts of the TP was influenced by both pre-season rainfall and pre-season temperature, and the start of the growing season was earlier and the end was delayed. The climate in the eastern plateau is warm and humid, and the vegetation type is mainly alpine meadow. Therefore, the increase of temperature will promote the growth of vegetation [41]. We suspect that the response of vegetation to air temperature change in the TP is greater than the influence of temperature change on vegetation, so the increase of vegetation does not reduce air temperature.

The influence of urban change on air temperature change is not significant in the TP, which may be due to the low level of urbanization and small dispersion of urban layout, resulting in insignificant human activities [42,43]. Although the growth rate of urban area on the TP has increased in recent years, it is far below that of eastern China.

Snow is very sensitive to climate change, especially temperature. In recent decades, the rising rate of temperature in the TP has been higher than the average temperature in the Northern Hemisphere, which has accelerated the snow melting and glacier retreat. At the same time, the increase of temperature also leads to the decrease of precipitation, which will further make snow and glaciers melt [44].

The insignificant influence of water cover on air temperature change in most seasons is due to the long-term low temperature in the TP, and some water bodies may contribute to temperature change in the form of lake ice. The effect of water cover on air temperature change in summer is of significance because the increase of water cover caused by melted snow and ice due to higher temperature in summer.

Although the impact of bare land on air temperature change has a single factor significance, it is not statistically significant. We suspect that bare land is restricted by a variety of land cover changes, while the changes of different land cover types are sometimes contrary, resulting in the irregular change of bare land.

## 5. Conclusions

In order to discover the relationship between air temperature change and its dynamic influencing factors (carbon dioxide concentration, vegetation cover, urban cover, snow cover, water cover, and bare land cover) in the eastern TP, a quantitative model in different periods (annual/seasonal) was established between the standardized temperature change and its dynamic influencing factors. The results showed that data standardization could not only remove the interference from static influencing factors (natural factors), but also highlight the variation characteristics of each dynamic parameter. The main conclusions are as follows.

(1) The 100 km buffer zone of meteorological station was used as the research object to conduct experimental research and obtained certain results.

(2) The main dynamic influencing factors influencing the temperature change in the eastern TP in different periods were discovered. The factors that significantly affect temperature were different in different periods.

(3) A model between the standardized temperature change and its dynamic influencing factors in the eastern TP was established.

In annum, the standardized snow cover is most correlated with the standardized temperature (40.4%—the ratio of absolute value of the coefficient of this factor to the sum of absolute values of the coefficient of all factors), followed by annual carbon dioxide concentration (35.7%) and vegetation cover (23.9%). In spring, the most correlated standardized factors were vegetation cover (48.4%) and snow cover (51.6%). In summer, the most correlated standardized factors were snow cover (41.7%), water cover (26.1%), carbon dioxide concentration (21.8%), and vegetation cover (10.4%). In autumn, the most correlated standardized factors were carbon dioxide concentration (43.9%), vegetation cover (36.0%), and snow cover (20.1%). In winter, the most correlated standardized factors were snow cover (60.8%) and vegetation cover (39.2%). In any period, the results showed that snow cover and water cover were negatively correlated, and the other factors were positively correlated.

The results showed that the corresponding correlation coefficients between the predicted values and the actual values of the model for annum, spring, summer, autumn, and winter were 0.451, 0.530, 0.516, 0.622, and 0.414, respectively. The standardized data of each dynamic influencing factor involved in modeling in each period passed the *t*-test. Therefore, the model has a good predictive ability, which not only provides a theoretical basis for the research of climate change in the eastern TP, but also points in the direction for establishing the relationship between temperature change and its dynamic influencing factors in similar areas.

## Figures and Tables

**Figure 1 sensors-22-06196-f001:**
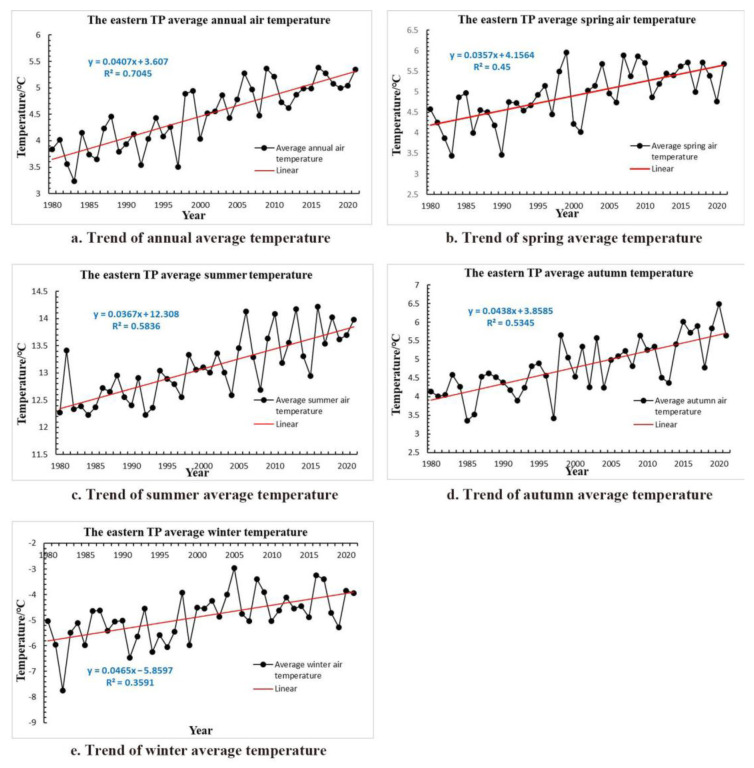
Trend of average air temperature over the eastern TP during 1980–2021 in (**a**) annum; (**b**) spring; (**c**) summer; (**d**) autumn; and (**e**) winter.

**Figure 2 sensors-22-06196-f002:**
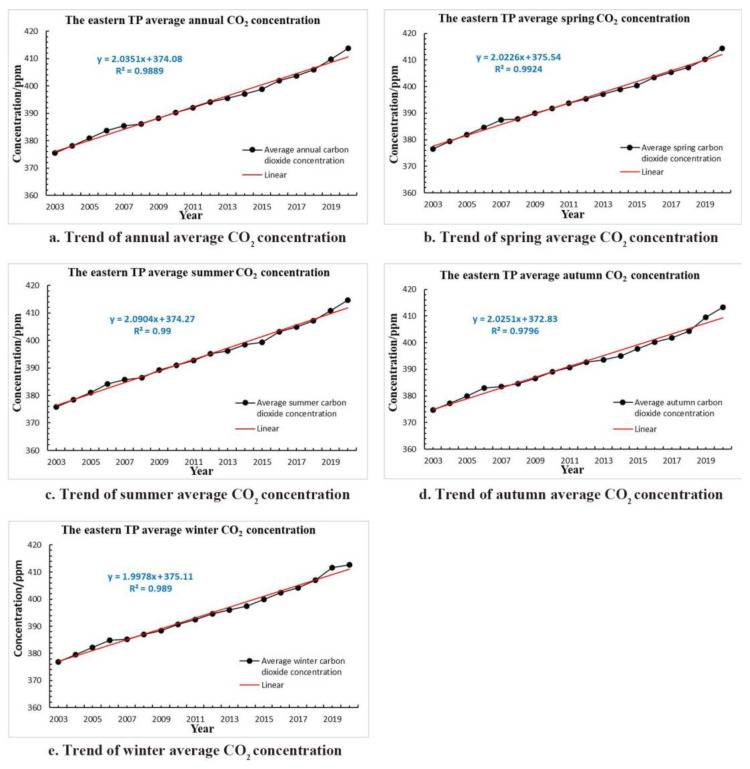
Trend of average CO_2_ concentration over the eastern TP during 2003–2020 in (**a**) annum; (**b**) spring; (**c**) summer; (**d**) autumn, and (**e**) winter.

**Figure 3 sensors-22-06196-f003:**
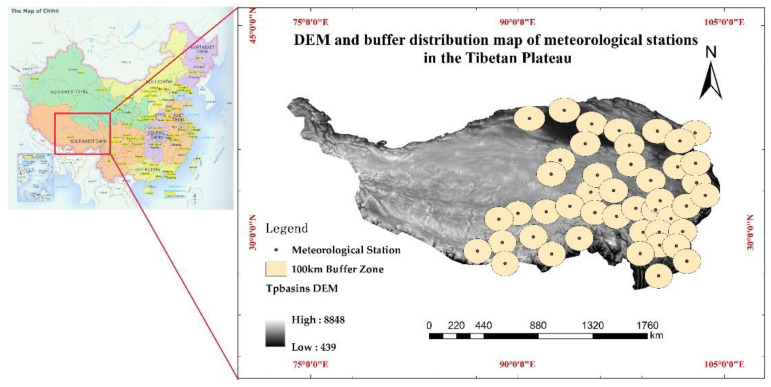
Map of the study area with meteorological stations (map on the left shows the location of study area in China).

**Figure 4 sensors-22-06196-f004:**
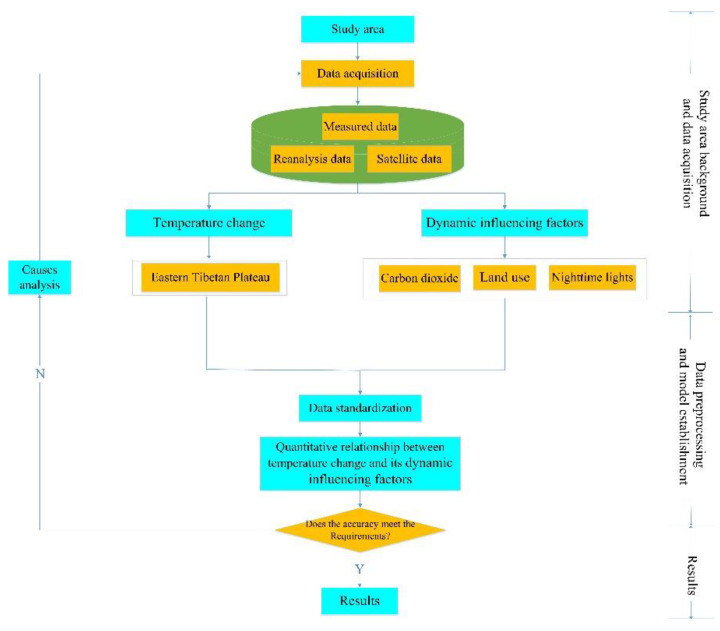
Technical flowchart for establishing the model between the standardized temperature change and dynamic influencing factors.

**Figure 5 sensors-22-06196-f005:**
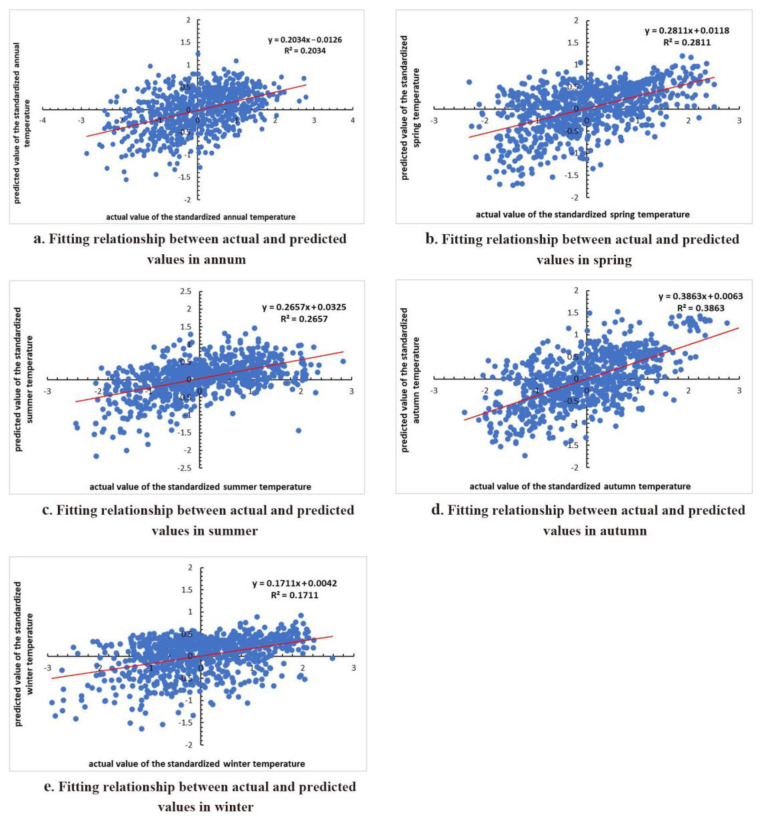
Model between temperature change and its dynamic influencing factors in different periods: (**a**) annum; (**b**) spring; (**c**) summer; (**d**) autumn, and (**e**) winter.

**Table 1 sensors-22-06196-t001:** Data acquisition for each data type.

Data Type	Temporal Resolution	Spatial Resolution	Data Source
Air temperature	Daily	Measured	NOAA National Center for Environmental Information (NCEI) (https://www.ncei.noaa.gov/) (accessed on 13 January 2022)
Carbon dioxide concentration	Monthly	0.75° × 0.75°	*CAMS global greenhouse gas reanalysis (EGG4) monthly averaged fields*, Copernicus Atmospheric Monitoring Service (https://ads.atmosphere.copernicus.eu/cdsapp#!/search) (accessed on 27 February 2022)
Vegetation cover	Monthly	0.011° × 0.011°	*MODIS/Terra Vegetation Indices Monthly L3 Global 1 Km SIN Grid* (MOD13A3 V061), LP DAAC Data Center (https://lpdaac.usgs.gov/) (accessed on 23 March 2022)
Nighttime light index products	Yearly	0.0083° × 0.0083°	*A Prolonged Artificial Nighttime-light Dataset of China (1984–2020)*, National Tibetan Plateau Data Center (https://data.tpdc.ac.cn/en/) (accessed on 29 March 2022) DOI: 10.11888/Socioeco.tpdc.271202
Snow cover	Daily	0.0053° × 0.0053°	*Daily cloud-free snow cover products for Tibetan Plateau from 2002 to 2021*, National Tibetan Plateau Data Center (https://data.tpdc.ac.cn/en/) (accessed on 28 March 2022) DOI: 10.11888/Cryos.tpdc.272204
Water cover	Monthly	0.011° × 0.011°	*MODIS/Terra Vegetation Indices Monthly L3 Global 1 Km SIN Grid* (MOD13A3 V061), LP DAAC Data Center (https://lpdaac.usgs.gov/) (accessed on 23 March 2022)
Bare land cover	Monthly	0.011° × 0.011°	*MODIS/Terra Vegetation Indices Monthly L3 Global 1 Km SIN Grid* (MOD13A3 V061), LP DAAC Data Center (https://lpdaac.usgs.gov/) (accessed on 23 March 2022)

**Table 2 sensors-22-06196-t002:** Correlation coefficients between the standardized temperature change and dynamic influencing factors in annum.

No.	Dynamic Influencing Factor	Correlation Coefficient
**1**	the standardized annual carbon dioxide concentration	0.305
**2**	the standardized annual vegetation cover	0.365
**3**	the standardized urban cover	0.132 (low)
**4**	the standardized annual snow cover	−0.320
**5**	the standardized annual water cover	Failed the correlation test
**6**	the standardized annual bare land cover	−0.335

**Table 3 sensors-22-06196-t003:** Correlation coefficients between the standardized temperature change and dynamic influencing factors in spring.

No.	Dynamic Influencing Factor	Correlation Coefficient
**1**	the standardized spring carbon dioxide concentration	Failed the correlation test
**2**	the standardized spring vegetation cover	0.465
**3**	the standardized urban cover	Failed the correlation test
**4**	the standardized spring snow cover	−0.466
**5**	the standardized spring water cover	0.151 (low)
**6**	the standardized spring bare land cover	−0.416

**Table 4 sensors-22-06196-t004:** Correlation coefficients between the standardized temperature change and dynamic influencing factors in summer.

No.	Dynamic Influencing Factor	Correlation Coefficient
**1**	the standardized summer carbon dioxide concentration	0.339
**2**	the standardized summer vegetation cover	0.312
**3**	the standardized urban cover	0.178 (low)
**4**	the standardized summer snow cover	−0.368
**5**	the standardized summer water cover	−0.310
**6**	the standardized summer bare land cover	−0.117 (low)

**Table 5 sensors-22-06196-t005:** Correlation coefficients between the standardized temperature change and dynamic influencing factors in autumn.

No.	Dynamic Influencing Factor	Correlation Coefficient
**1**	the standardized autumn carbon dioxide concentration	0.462
**2**	the standardized autumn vegetation cover	0.524
**3**	the standardized urban cover	0.279 (low)
**4**	the standardized autumn snow cover	−0.376
**5**	the standardized autumn water cover	0.235 (low)
**6**	the standardized autumn bare land cover	−0.421

**Table 6 sensors-22-06196-t006:** Correlation coefficients between the standardized temperature change and dynamic influencing factors in winter.

No.	Dynamic Influencing Factor	Correlation Coefficient
**1**	the standardized winter carbon dioxide concentration	Failed the correlation test
**2**	the standardized winter vegetation cover	0.360
**3**	the standardized urban cover	−0.094 (low)
**4**	the standardized winter snow cover	−0.392
**5**	the standardized winter water cover	0.190 (low)
**6**	the standardized winter bare land cover	−0.312

**Table 7 sensors-22-06196-t007:** *T*-test for each parameter (annual).

	Coefficients	*p*-Value	Lower 95%	Upper 95%
Intercept	−0.0279	0.378	−0.0899	0.0341
The standardized annual carbon dioxide concentration	0.226	$7.674\times 10^{-10}$	0.155	0.297
The standardized annual vegetation cover	0.151	$3.042\times 10^{-4}$	0.0693	0.233
The standardized annual snow cover	−0.256	$6.173\times 10^{-11}$	−0.332	−0.180

**Table 8 sensors-22-06196-t008:** *T*-test for each parameter (spring).

	Coefficients	*p*-Value	Lower 95%	Upper 95%
Intercept	−0.0351	0.9062	−0.0620	0.0550
The standardized spring vegetation cover	0.303	$2.024\times 10^{-16}$	0.232	0.373
The standardized spring snow cover	−0.323	$1.073\times 10^{-16}$	−0.398	−0.248

**Table 9 sensors-22-06196-t009:** *T*-test for each parameter (summer).

	Coefficients	*p*-Value	Lower 95%	Upper 95%
Intercept	0.00565	0.853	−0.0540	0.0653
The standardized summer carbon dioxide concentration	0.193	$9.236\times 10^{-8}$	0.123	0.263
The standardized summer vegetation cover	0.0924	0.0258	0.0112	0.174
The standardized summer snow cover	−0.369	$1.096\times 10^{-18}$	−0.448	−0.289
The standardized summer water cover	−0.231	$1.722\times 10^{-10}$	−0.301	−0.161

**Table 10 sensors-22-06196-t010:** *T*-test for each parameter (autumn).

	Coefficients	*p*-Value	Lower 95%	Upper 95%
Intercept	0.0138	0.616	−0.0401	0.0677
The standardized autumn carbon dioxide concentration	0.358	$7.24\times 10^{-30}$	0.299	0.418
The standardized autumn vegetation cover	0.293	$2.23\times 10^{-12}$	0.213	0.374
The standardized autumn snow cover	−0.164	$4.13\times 10^{-5}$	−0.242	−0.0858

**Table 11 sensors-22-06196-t011:** *T*-test for each parameter (winter).

	Coefficients	*p*-Value	Lower 95%	Upper 95%
Intercept	0.000236	0.994	−0.0636	0.0640
The standardized winter vegetation cover	0.184	$4.071\times 10^{-5}$	0.0966	0.272
The standardized winter snow cover	−0.285	$4.473\times 10^{-10}$	−0.373	−0.196

## Data Availability

Not applicable.

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
