# Peer review of "A Model between Near-Surface Air Temperature Change and Dynamic Influencing Factors in the Eastern Tibetan Plateau, China"

_sensors, 2022, doi:10.3390/s22166196_

Round 1

Reviewer 1 Report

In this manuscript, the multi-regression model between the air temperature and the most influencing factors is established. While such a model might have kind of predictive value, the quality of presentation of the problem, methodology, results and a reference to other research of similar type is rather low. Therefore, it is adviced to correct manuscript making it more comprehensive. Introduction should mention similar research, and not describe what's up in world ocean...Scientific questions, hypotheses and novelty should be mentioned. As it is mentioned that ".. the highest altitude of the meteorological station is about 4720m, and the lowest is about 2380m", it should be indicated if the model is suitable only for such altitude within considered region, or it is considered to be suitable for the whole region with altitudes ranging from 440m to 8500m a.s.l. By the way, proper units m a.s.l. should be added. Section 2.2.2 should be more comprehensive. From the Result section, please remove all methodology description. Finally, the Discussion section should be more synthetic, without so many numbered points. Generally, the manuscript should be carefully and thoroughly improved and synthetized to make it more clear. The proofreading is a must, as several parts of the text are not understandable.

Title: please specify that this concerns air temperature, namely Near-Surface Air Temperature

Line 33: instead WTO, should be WMO

Line 39: "For example, by the end of the 21st century, glaciers have 39 lost 18-36% of the ice volume in 2015 [1]." No sense in this sentence. Besides, Line 430: wrongly writtern reference: Trevor is the first name, Letcher should be cited. The same with the names of the Authors, please correct. You abbreviated surnames insted of first names.... Regine and Mathias are first names, and these should be abbreviated, and surnames should be given. Please pay such attention to the rest of literature positions.

Line 41: "The evaporation of rock and soil moisture" ?? probably wrong translation

Lines 45-48: this is not he subject of this article

Line 54: please cite proper literature

Line 65:"The temperature in this paper refers to the air average temperature." It does not explain what is meant by this term. Plese specify. Besides, please specify research questions and hypotheses investigated in this reseach. Adding the novelty would explain your contribution to the science.

Lines 87: Figure 1: text is not visible

Lines 80-82: "Although the average annual temperature 80 of the Qinghai-Tibet Plateau is relatively low, it has shown a rising trend in recent decades, 81 from about 3.8℃ in 1980 to about 5.3℃ today." please cite the source of this information

Line 106: Table 1: please provide versions of data and appropriate links

Line 109: "data with Resolution of 40 (Cg-XII)..." this not explain the resoultion of data; which resolution, spatial, temporal???

Line 115: "a gridded level 3 product after sinusoidal projection"; it is not understandable, sorry

Lines 130-132: "According to the daily surface meteorological observations of 45 meteorological stations in the eastern Qinghai-Tibet Plateau from 2003 to 2020, the monthly data of the stations were obtained by statistical sorting, and the annual (seasonal) data were obtained." itdoes not explain the procedure...

Line 149: "The standardized data is obtained by dividing anomaly to its average."  please do not repeat such information, below is more detailed description, so please make it more comprehensive

Line 240-241: "between the standardized temperature and its dynamic influencing factors."  But in lines 60-61, there is a message that you consider temperaure change and not temperature : "Therefore, establishing a quantitative model between temperature change and its dynamic influencing factors". Also, in Figure 2 You indicate that You consider temperature change and not temperature. So please clear it up in all places in the text. Lines 293-294 "Therefore, the temperature change (standardized autumn temperature)..." please correct if you mean the temperature or temperature change. These two terms have completely different meaning.

Section RESULTS: please do not repeat the methodology here. Important information on methods should be moved to the previous section.

Lines 319-320: please explain what is meant by: "the vegetation cover of the Qinghai-Tibet Plateau has increased in different seasons"; by what means it has increased??

Line 322: "...:every 0.1 increase in vegetation cover"; 0.1 of what???

Reviewer 2 Report

The research applies the least squares multivariate regression in order to investigate factors that impact temperature change in eastern Qinghai-Tibet Plateau using 18 years long data from observations and satellite measurements. Although the research is interesting, I am concerned with some decisions within the methodology that should be corrected. The most important, the analysis should be done and analyzed for all seasons, not only for the autumn.

Major comments: 1. Since the measurements of the temperature are available only in the eastern part of the Plateau, why the authors did not use any available temperature reanalysis dataset in order to cover the entire Plateau?

2. The authors should show the changes in temperature across the seasons and stations. Does the temperature in autumn (and other seasons) rises or falls?

3. It would be beneficial to graphically present values of the impacting factors across the stations. It would be nice to see how much they differ among the stations (and their buffer zones). Especially the C02 concentration.

4. My main concern is why the multivariate regression is not done for each season? Why the season selection is done using only correlation with CO2 concentration? Why this way of the selection makes it reliable? Why the authors did not look at correlations with every factor in every season? My opinion is that the analysis has to be done for each season, and results explain better.

5. How seasonal data on water cover and bare land were extracted from the annual frequency of the original datasets used? Bare land surfaces are changing over the year, together with the vegetation cover (especially in the agricultural areas). Water bodies are probably frozen during the cold months and become ice.

Minor comments:

Lines 14/15: Causes of modern climate change are already very well known and proved.

Line 22: “CO2 concentration was the closest factor to temperature change” – word closest is not appropriate in this context, please rephrase.

Line 33: Abbreviation is WMO, not WTO.

Lines 32/34: A verb is missing at the end of this sentence.

Lines 39/40: The sentence is not phrased well. If it is about the change that has already happen it should be 20th century. If it is about future change, the sentence should be in the future tense.

Lines 45/49: Not sure how changes in the ocean are directly linked to the changes at the Plateau. Instead, it would be beneficial to describe changes in snow precipitation (amount, timing, etc.) and its broad impact to hydrological cycle.

Lines 50/53: Dynamics of the atmosphere is one of the most important dynamic factor that influences the temperature, among other meteorological parameters.

Line 65: air average temperature – average of what? Daily, monthly, seasonal, annual...? Please specify.

Line 94/95: “CO2 concentration is...” – this sentence is redundant.

Lines 93/101: Sentences here are pretty much pointless or state something that is common knowledge. Instead, it would be beneficial to briefly explain how each of the selected factors influences the air temperature. That way the authors would justify the selection of the factors in the research.

Line 132: please specify the seasons – which months are included in which season?

Lines 308/310: the sentence is redundant. These changes have already happen and will continue.

Lines 311: please explain which seasonal effects. Please show the seasonal difference in CO2 concentration.

Line 314: Sun, Earth Line 315: space

Lines 328/330: sentence not clear, transfer of what?

Lines 319/330: Not sure if the conclusions here are well derived. The vegetation cover in autumn is larger due to increase in temperature (longer vegetation season).

Round 2

Reviewer 1 Report

Dear Authors: Only small improvement to be considered:

Line 129: Figure 2?

Figure 3: Please enlarge red labels (geographical coordinates)

Reviewer 2 Report

The authors responded adequately to all of my comments. The methodology is changed according to the suggestions, which led to more comprehensive and meaningful results.

Minor English corrections are still needed.
